# Differences in Physico-Chemical Properties and Immunological Response in Nanosimilar Complex Drugs: The Case of Liposomal Doxorubicin

**DOI:** 10.3390/ijms241713612

**Published:** 2023-09-02

**Authors:** Dorelia Lipsa, Davide Magrì, Giacomo Della Camera, Rita La Spina, Claudia Cella, Irantzu Garmendia-Aguirre, Dora Mehn, Ana Ruiz-Moreno, Francesco Fumagalli, Luigi Calzolai, Sabrina Gioria

**Affiliations:** 1European Commission, Joint Research Centre (JRC), 21027 Ispra, Italy; ldoreliasimona@yahoo.com (D.L.); davide.magri.mail@gmail.com (D.M.); giacomo.d.c.88@gmail.com (G.D.C.); ritalaspina.rls@googlemail.com (R.L.S.); claudia.cella@ec.europa.eu (C.C.); irantzu.garmendia-aguirre@ec.europa.eu (I.G.-A.); dora.mehn@ec.europa.eu (D.M.); ana.ruiz-moreno@ec.europa.eu (A.R.-M.); francesco-sirio.fumagalli@ec.europa.eu (F.F.); luigi.calzolai@ec.europa.eu (L.C.); 2Institute of Biochemistry and Cell Biology (IBBC), National Research Council (CNR), 80131 Naples, Italy

**Keywords:** Doxil, doxorubicin, safety assessment, nanomedicines, particle size distribution, liposome, endotoxin, inflammation, complement activation, cytokines

## Abstract

This study aims to highlight the impact of physicochemical properties on the behaviour of nanopharmaceuticals and how much carrier structure and physiochemical characteristics weigh on the effects of a formulation. For this purpose, two commercially available nanosimilar formulations of Doxil and their respective carriers were compared as a case study. Although the two formulations were “similar”, we detected different toxicological effects (profiles) in terms of in vitro toxicity and immunological responses at the level of cytokines release and complement activation (iC3b fragment), that could be correlated with the differences in the physicochemical properties of the formulations. Shedding light on nanosimilar key quality attributes of liposome-based materials and the need for an accurate characterization, including investigation of the immunological effects, is of fundamental importance considering their great potential as delivery system for drugs, genes, or vaccines and the growing market demand.

## 1. Introduction

Liposomes are microscopic spheres, which were developed as drug-delivery vehicles/systems in the 1980s. The first liposome-based pharmaceuticals were approved for commercial use in the 1990s. Since then, the liposome technology has provided intelligent solutions to solve challenges in pharmacology, such as increasing drug solubility, reducing drug toxicity, and improving targeted drug release [1,2]. Liposomes have three distinct compartments that can be used to carry various compounds such as, e.g., drugs: the interior aqueous compartment; the hydrophobic bilayer; and the polar inter-phase of the inner and outer leaflet [3,4,5].

Depending on the chemical nature of the compound to be encapsulated it will be localised to one or other of the compartments. For this, several liposome–drug formulations have been successfully approved and used in clinics over the past two decades [3].

Water-soluble drugs tend to be localised in the aqueous compartment, encapsulated in the liposome, such as doxorubicin (Doxil), doxorubicin (Myocet), and daunorubicin (DaunoXone), or intercalated in the liposome membrane, such as amphotericin B (AmBisome), amphotericin (Albelcet B), benzoporphyrin (Visudyne), or muramyltripeptide-phosphatidylethanolamine (Junovan).

In addition, in response to the growing market demand for these products, the involvement of third-party manufacturers is inevitable and, in the case of patent expiration, as happened for the liposomal doxorubicin Doxil (original patent expired in 2009), several companies have embarked in the development and production of so-called nanosimilar products [6].

These nanosized systems, unlike classic small-molecule drugs, are sophisticated multicomponent formulations well known as complex non-biological complex drugs (NBCDs); therefore, the conventional qualitative assessment approach, such as the bioequivalence criterion applied in biosimilars, cannot be applied [7,8]. For this reason, the debate on nanosimilars and the definition of key quality attributes and evaluation methods for their regulatory management remains open [9,10]. Doxil is the emblematic example in the race for nanosimilars, demonstrated by the numerous manufacturers offering doxorubicin liposomal formulations on the market.

The efficacy and biological fate of these complex nanoformulations depend on the combination of physicochemical properties, such as the composition of the carrier and the internal load, the size of the liposome, and the surface charge and morphology. However, the concentrations of particles, average size, composition and content of encapsulated active substances are just some of the quality requirements necessary for the development of an NBCD [11]. These parameters are very sensitive to synthesis conditions and the hydrodynamic behaviour shown under these conditions by their key ingredients—among them poly(ethylene glycol)-lipids (PEG-lipids) [12].

Even in the case of the ostensibly identical composition of liposomal medicinal products, variation in the production and product/process control technology can lead to products with different therapeutic performance. Therefore, the complete characterization of their physicochemical properties, stability, and pharmacokinetics is critical to establish their safe and effective use [13].

Considering the increasing number of liposome applications, several guidelines were released by regulatory agencies, sharing their experience in the scientific evaluation of liposome formulations. The European Medicinal Agency (EMA) has published a “Reflection paper on the data requirements for intravenous liposomal products developed with reference to an innovator liposomal product” in order to provide general recommendations on the data requirements to demonstrate comparability of liposomal formulations [14]. For doxorubicin hydrochloride, the Food and Drug Administration (FDA) has released a specific draft guidance according to which, to be eligible for bioequivalence studies, the test product should have equivalent characteristics including liposome composition, the state of the encapsulated drug, the internal environment of liposomes, liposome size distribution, number of lamellae, grafted PEG at the liposome surface, electrical surface potential or charge, and in vitro leakage rates compared to the reference standard [15].

Considerations regarding safety and drug efficacy also require that liposome formulations maintain their properties, i.e., remain stable, from the time of preparation until administration.

Many of these factors affect the biological fate and the bio-distribution of the NBCD and the formulations should be designed to prevent their immunogenicity, an issue that can play serious roles in ‘nanosimilar’ biological effects. This is a particularly important point to consider for formulations injected into the bloodstream, directly exposed to adaptive immunity [16]. The pre-existing anti-PEG immunity of the population reported in more studies calls for careful attention to be paid while developing these technologies [17,18]. It should be also considered that the interactions with the complement system might lead to recognition of defects with even minimal variations in the surface properties, shape, and morphology of nano-objects. Thus, physico-chemical differences might contribute to trigger complement reactions [19,20,21]. In this work, we compared two formulations of Doxil nanosimilars produced by different manufacturers as a case study. Although the two formulations are considered nanosimilars, we observed differences in the physico-chemical characteristics that could explain the different toxicological effects (profiles) measured in human cells and serum exposed to the selected formulations. Focusing on in vitro toxicity and on the immunological effects at the level of cytokines production and complement activation (iC3b fragment), we highlighted differences in the immune response induced by the NBCD as a whole, attributing some effects to the free drug, but also recognizing the fundamental role of the carrier.

## 2. Results

Two commercially available formulations of Doxil, namely Dox1 and Dox2, and their corresponding carriers (Dox1C and Dox2C), were selected as representative examples. The applied multi-step characterization strategy is illustrated in Figure 1.

### 2.1. Physic-Chemical Characterization of Liposomes

Batch-mode dynamic light scattering (DLS) was used as preliminary screening tool to evaluate the particle size distribution (PSD) and the polydispersity of the samples. The data are summarized in Table 1 and illustrated in Figure 2A. Batch-mode DLS measurements were repeated during the duration of the whole study to monitor the stability of the selected samples. They proved to be stable for the whole duration of the study.

Dox1 and its corresponding carrier (Dox1C) showed very similar size with a hydrodynamic diameter (zeta average) of about 77 nm. Their polydispersity index is also low, and they show a quite narrow size distribution, suggesting that the products are homogeneous. On the other hand, Dox2 and its corresponding carrier (Dox2C) have wider size distribution with PDI values exceeding 0.1 in both cases and zeta average values of 107 and 116 nm, respectively.

### 2.2. AF4-DLS Analysis

Batch mode DLS is known to have intrinsic limitations in the analysis of polydisperse suspensions because of the stronger light scattering of larger particles that might outweigh the signal of smaller ones. In order to overcome this limitation, DLS can be used in flow mode, coupled inline to a size separation instrument, such as field flow fractionation techniques. In our case, the liposomal doxorubicin formulations were also tested by asymmetric flow field flow fractionation coupled with DLS (AF4-DLS). The data are illustrated in Figure 2B–D.

As shown in Figure 2B, the two liposomal doxorubicin formulations exit from the AF4 column with different elution profiles. Dox1 and its corresponding carrier both appear as a single particle population with particles in the size range of 66–150 nm with D_FWHM_ of 87 and 84 for Dox1 and Dox1C, respectively (Figure 2C). Dox2 and the non-loaded carrier elute showing two overlapping peaks in the light-scattering signal due to (at least) two contributing particle populations: smaller particles with a hydrodynamic diameter of about 40–70 nm (D_FWHM_ of 51 and 52 for Dox2 and Dox2C) and larger ones in the 70–150 nm range D_FWHM_ of 93 and 111 for Dox2 and Dox2C, respectively (Figure 2D). The presence of different-sized particle populations in Dox2 and Dox2C explains the wider peaks detected in batch mode DLS and the larger polydispersity index values.

### 2.3. AUC Confirms Results Obtained with AF4-DLS

The two loaded formulations and their carriers were also characterized by analytical ultracentrifugation (AUC) to assess their size distribution with a true orthogonal technique. Sedimentation coefficient distributions were calculated applying the ls-g*(s) model of the Sedfit [22,23,24] and transformed to Stokes diameter distributions, using estimated liposome densities of 1.055 g/mL for loaded liposomes and 1.045 g/mL for control carrier particles determined in former studies [25]. Results are illustrated in Figure 3.

The presence of a monodispersed particle population in the case of Dox1 and its corresponding carrier were confirmed by the AUC measurements showing Stokes diameter (distribution mode) of about 80 nm for Dox1, and about 60 nm for Dox1C and suggesting slightly lower diameter for the “empty” liposomes. The AUC-calculated size distribution showed much lower sizes for Dox2 and its non-loaded carrier, with Stokes diameter modes at about 40 nm and additional particle population(s) at higher sizes—fully supporting the AF4-DLS results. Stokes diameter results for the monodispersed Dox1 and Dox1C samples are very close to the ones obtained by DLS batch mode and AF4-DLS. While batch mode DLS suggested only broader size distribution in the case of Dox2 and Dox2C, both AUC and AF4-DLS were able to resolve the size distribution of the more polydispersed samples and revealed that Dox2 and Dox2C contain large amount of particles that have unexpectedly low diameters.

Moreover, AUC data collected at 514 nm in the case of loaded formulations reveal another important difference linked to the presence of a small species that is not sedimenting at 10,000 rpm rotational speed at a detectable velocity (Figure 4). This signal belongs to the free doxorubicin hydrochloride (DoxHCl). While the intensity of this signal, which does not change after the sedimentation of all particles, is practically 0 in the case of Dox1, for Dox2 this contribution is much higher and indicates that the formulation contains about 13% of free doxorubicin hydrochloride outside the carrier.

### 2.4. Determination of Endotoxin Content

Pyrogenic material can affect the biological evaluation data, therefore we measured the endotoxin contamination level of both Doxil formulations and their respective carriers by running a kinetic chromogenic limulus amoebocyte lysate (LAL) assay. Results are reported in Table 2.

Endotoxin screening shows that Dox1 formulation and its respective carrier do not contain any traces of endotoxin, whereas Dox2 formulation and its carrier have measurable endotoxin contamination. In particular, this corresponds to an endotoxin level of 4.10 and 3.01 EU/mL for Dox2 and Dox2C, respectively. To eliminate the risk of possible interferences of the formulation with the optical-based LAL assay, spike samples have been included and provided a good recovery rate.

### 2.5. Cytotoxicity Assessment

The viability/metabolic activity of LLCPK1 cells exposed to each Doxil formulation or its respective carrier were assessed by 3-(4,5-dimethylthiazol-2-yl)-2,5-diphenyl-2H tetrazolium bromide (MTT) assay, after 48 h of exposure. Free doxorubicin (DoxHCl) was also included for comparison.

Results of the MTT assay are presented in Figure 5. Data are in agreement with morphological observations. Significant differences between the two Doxil formulations are visible in the MTT assay (*p* < 0.05). Interestingly, an IC50 value of about ten times lower was obtained in the case of DoxHCL compared to Dox1 (0.3 vs. 3 µg/mL, respectively). No significant differences are observed between the effect of the Dox1 formulation and the respective Dox1C control, while Dox2 appears to be more harmful for the cells than its non-loaded control carrier (*p* < 0.01).

### 2.6. Effects of Doxil Formulations on Complement Activation

The immunotoxicity potential of the two Doxil formulations and their corresponding carriers was investigated on commercially available human pooled serum. The selected materials were tested at the final concentration of 200 µg/mL of the liposomal formulation; results are reported in Figure 6 and show that the iC3b levels for Dox1 and Dox2 were 2.1- and 3.3-fold higher with respect to the negative control, whereas serum incubated with the respective carriers caused a 1.2- and 1.5-fold increase for Dox1C and Dox2C, respectively.

For comparison, the capacity of free doxorubicin hydrochloride (DoxHCl) to raise the levels of iC3b was also evaluated by exposing human pooled serum to DoxHCl concentrations ranging from 2.5 to 20% of the total theoretical doxorubicin load of the liposomes (at the applied 200 µg/mL doxorubicin concentration). As shown in Figure 7, the amount of iC3b complement fragment measured in human serum exposed to free doxorubicin at the lowest concentration tested (2.5%) was comparable to the negative control (human serum incubated with PBS). Increased iC3b levels were observed when sera were treated with higher concentration of doxorubicin, e.g., 20% free doxorubicin induced a 1.5-fold increase in the amount of iC3b with respect to the negative control.

### 2.7. Profile of Cytokines/Chemokines of Human Serum Exposed to Doxil Formulations

The amount of cytokines/chemokines in human pooled serum treated with the two formulations of Doxil, their respective carriers and free doxorubicin was also assessed. As reported in Figure 8, a significant amount of cytokines/chemokines were raised by Dox2; in particular TNF-α, IL-6, IL-8, GM-CSF, IFN-γ, IL-10 and IL-4 (*p* < 0.001); and IL-2 (*p* < 0.01), whereas no statistically significant amount of IL-1β and IL-1ra was detected.

In comparison, in the samples treated with Dox1 an increased amount of TNF-α, IL-6, IL-8, GM-CSF and IL-4 (*p* < 0.05), IFN-γ (*p* < 0.01) was determined whereas no variations in the levels of IL-2, IL-10, IL-1β and IL-1ra were measured. When human serum was treated with Dox2C, significant amounts of TNF-α (*p* < 0.05) were found; however, no other inflammatory markers analysed were significantly modified with respect to the control. Dox1C had no significant effect on any of the cytokine/chemokine levels.

Interestingly, significant levels of TNF-α (*p* < 0.001) were found in the samples treated with 10 and 20% free doxorubicin. On one hand, free doxorubicin at a concentration of 20% (of the total load) did not have a statistically significant impact on IL-6, IL-8, GM-CSF, IL-4, IL-1β, and IL-1ra, whereas a significant amount of IFN-γ (*p* < 0.01), IL-2 (*p* < 0.05), and IL-10 (*p* < 0.001) was determined. On the other hand, in the serum samples exposed to 10% free doxorubicin, only TNF-α (*p* < 0.001) and IL-10 amounts were detected at significant levels compared to the negative control (*p* < 0.01).

Figure 9 provides an overview of the concentration profiles of cytokines/chemokines measured in human serum exposed to the different selected materials: the two formulations of Doxil, their respective carriers and free doxorubicin (10 and 20% of theoretical load).

## 3. Discussion

The main purpose of this work was to highlight the importance of the physico-chemical properties in the safety of nanoformulations. At present, liposomes are used as carriers of chemotherapeutics due to their biocompatibility and versatility as demonstrated by the number of successfully approved formulations by FDA and EMA [26]. In the last decade, much attention has been dedicated to assessing the impact of nanocarriers [3,27,28,29] with respect to the whole range of nanodrugs in terms of immune and inflammatory response [20,30,31]. In this work, in vitro toxicity and the immunological effects at the level of complement activation (iC3b) and cytokine/chemokine production of two nanosimilar formulations of Doxil and their corresponding carriers were tested. Free doxorubicin hydrochloride was also included for comparison. Our results confirm that physico-chemical properties should be correlated with in vitro and ex vivo toxicological effects of nano-drugs and reported prior to nano-drug administration to avoid any adverse effects, including complement-activation-related pseudoallergy (CARPA), hypersensitivity, and inflammasome activation. In our case, two formulations of Doxil produced by different manufacturers, considered nanosimilars, presented different toxicological profiles, which can be attributed to the diverse physico-chemical characteristics of the formulations. When comparing the two formulations, at first, we characterized the particle size and size distribution (PSD), a critical quality attribute, as the whole pharmacokinetic process of liposomes is dimension-dependent [32]. Size measurements were performed using orthogonal techniques as batch-mode DLS, an AF4-separation coupled sizing detector, and AUC, as previously suggested also by other authors for in-depth size characterization of nanomedicine products [33,34,35,36]. By applying batch-mode DLS to the samples under investigation, we identified differences between the two formulations, with Dox1 and its corresponding carrier, Dox1C, having similar size and being monodispered, whereas Dox2 and its corresponding carrier, Dox2C, possessing a wider size distribution and higher polydispersity.

In order to improve the performance of DLS measurements, we coupled the instrument to a field flow fractionation system (AF4). In the AF4 channel, the particle separation depends not only on size but also on surface properties [37,38,39]. Because of this, the calculation of hydrodynamic diameters from elution times is not possible without size calibrants with the same surface properties. However, the inline-coupled DLS reports the zeta-average size of the exiting particles, and in this setup weak-scatterer small particles and strong-scatterer large ones arrive at the DLS detector separated in time. This way, the measurement of the hydrodynamic diameter is expected to be more accurate. On one hand, AF4-DLS measurements confirmed the particle size results of the monodispersed products (Dox1 and Dox1C) and D_FWHM_ were found to be very close to the average size expected for Doxil nanosimilars based on the properties of Doxil (85 nm) [40]. On the other hand, AF4-DLS also indicated the more complex composition of Dox2 and Dox2C; in our case, it revealed the presence of a smaller, 40–50 nm particle fraction.

Each loaded Doxil nanosimilar was also analysed by AUC and compared to the respective empty formulation to confirm the findings of AF4-DLS and evaluate drug encapsulation efficiency, another critical quality attribute [14]. AUC was shown to be a powerful technique for the characterization of empty or drug-loaded liposomes [25]. On one hand, interference-based AUC measurements provide differential mass-based size distributions after the transformation of sedimentation coefficient distributions by considering the density (and shape) of particles, and the viscosity and density of the liquid phase. Thanks to the linear correlation between particle concentration and refractive index change (independent of particle size), interference-based measurements allow us to avoid the transformations necessary for absorbance-based measurements in the size range where Mie scattering becomes substantial [41]. On the other hand, AUC analysis with absorbance optics allows the estimation of free vs. loaded drugs if the active pharmaceutical ingredient (API) has a specific absorption wavelength, as in the case of doxorubicin. AUC, a true orthogonal method to light-scattering-based sizing techniques, in general confirmed the size characterization results of AF4-DLS. The slightly smaller diameters determined for non-loaded carrier(s) might be most probably explained by the overestimation of their density. Nevertheless, the presence of a much smaller size particle fraction in the polydisperse Dox2 and Dox2C samples was detected also by AUC.

Our findings confirm the conclusions suggested also by other authors: while batch-DLS-based hydrodynamic size results might be in good correlation with size information gained by coupling DLS to a size fractionation system (AF4) or size distributions from AUC measurements, AF4-DLS and AUC appear to provide more reliable size evaluation for heterodispersed systems [34].

As suggested also by other authors, AUC measurements are able to provide information not only on the size and density but also on the encapsulated vs. free drug ratio [42]. Our absorbance-based AUC measurements at the specific absorbance wavelength of the API revealed the presence of about 13% free drug outside the Dox2 liposomes. The amount of free drug in nanoformulations is a critical quality attribute, as the encapsulation changes the absorption, distribution, metabolism, and excretion (ADME) properties of the drug. In case of RNA/lipid nanoparticle formulations, the particles protect their cargo from RNAases, while in the case of Doxil, the liposomal encapsulation is supposed to protect the cells in general from the non-specific cytotoxic effect of the drug and dramatically decrease the systemic toxicity of doxorubicin hydrochloride. Typical free drug content is expected to be less than 2% in intact Doxil formulations [25]. Thus, 13% released drug together with a smaller than expected particle size suggest improper synthesis conditions or possible damage due to storage or shipment conditions.

The evaluation of the possible bacterial and endotoxin contamination (i.e., their sterility and pyrogenicity) is also an essential step in the nanomaterial characterization [28]. Our findings provide evidence that all samples were free from bacterial contaminations, whereas evaluation of the endotoxin contamination showed a measurable contamination in the case of Dox2, which is already present in its corresponding carrier, Dox2C.

The observed differences at the physico-chemical level between the two formulations influenced the interactions of these nanomaterials with the human cells and serum leading to different toxicity and immunological effects.

Focusing on in vitro toxicity and on the ex vivo immunological effects—in particular, on the complement activation in terms of quantifying the iC3b fragment levels and on the inflammatory response detected by the measurement of selected cytokines/chemokines—we indeed highlighted important differences between the two NBCDs and recognized also the fundamental role of the nanopharmaceutical carrier. Liposomes are well-known delivery systems [3,43,44,45,46] for, e.g., anti-cancer drugs, and received attention recently as vectors for SARS-CoV-2 vaccines. Nevertheless, tiny changes of their physico-chemical properties can trigger adverse effects such as CARPA [47]. It is well known that liposomes, even unloaded carriers, interact with the immune system by activating the most important, evolutionarily most conserved, rapid innate and inflammatory defensive line: the complement system [31,48]. Activation of the complement system by various foreign particles, including PEG-ylated liposomal formulations, may be transient, causing temporary stress but no permanent damage to the host. On the other hand, long-term or repeated activation may pose a significant health threat [49,50].

On one hand, liposomes are instrumental for the implementation of a paradigm shift in health care; on the other hand, however, safety issues require assessing the individual immune system response [27]. Therefore, to assess the safety of liposomes, accurate characterization at multiple levels is crucial. At the biological level, in vitro cytotoxicity testing is a fundamental part of the workflow assessment, starting with the identification of the hazardous potential of NBCD in the early phase of drug development [11,51]. To carry this out, there are several guidelines and standards that have been developed by OECD, ISO, and the US FDA and ASTM to evaluate the safety of nanoparticulates [52,53]. Therefore, both Doxil formulations and the respective carriers were assessed for direct cytotoxicity in vitro on LLCPK1 cells using a standard protocol [53]. Our results show differences in terms of reduction in the number of viable (metabolically active) cells between the two formulations, with Dox2 being more toxic than Dox1. This difference is not unexpected in the light of the free cytotoxic drug concentration detected by AUC. However, in our case the presence of the free active ingredient is combined also with abnormal particle morphology and endotoxin contamination. Interestingly, although the Dox2 and its corresponding carrier (Dox2C) showed similar size distributions and endotoxin contamination levels, their effects on cytokines were still divergent. As one of the main differences between these two samples is the presence of free doxorubicin in Dox2, control measurements applying similar concentrations of the free drug (10 and 20% of the total theoretical load) were also performed to separate the two effects. Likewise, we observed an IC50 value of about one order of magnitude lower in the case of DoxHCL than Dox1. Data confirmed that—even if the mechanism is unknown—the presence of free doxorubicin hydrochloride itself might affect various cytokines and chemokines concentrations in blood serum and most probably contribute to the complement-activation process by raising iC3b levels.

In our complement-activation experiments, using commercial human serum, we observed that both formulations at a concentration of 200 µg/mL (doxorubicin) activate the complement cascade, although with different potencies, with Dox2 > Dox1 (3.3- and 2.1-fold change, respectively). A small increase was also observed for their respective carriers (1.2- and 1.5-fold change for Dox1C and Dox2C, respectively), but more surprisingly, doxorubicin hydrochloride at similar concentration as the free drug in Dox2 was also able to raise the iC3b concentration in serum, thus contributing to trigger complement activation. iC3b is a breakdown product of the complement protein C3 and it may be seen in a number of different conditions, including inflammatory diseases (e.g., rheumatoid arthritis), autoimmune disorders (e.g., systemic lupus erythematosus) [54,55]. Overall, high levels of iC3b fragment can be a biomarker indicating a strong immune response against invading pathogens [54,55,56,57] or other underlying complement-mediated disorders contributing to tissue damage and inflammation [58,59,60]. Therefore, the iC3b data interpretation must be performed considering the clinical scenario—commercial pooled serum from healthy individuals in our case vs. serum from patients with preexisting pathological conditions.

Besides testing the direct toxicity of the nanosimilars on LLCPK1 cells and their potential to trigger immune response by activating the complement cascade, we studied the capacity of the nanosimilar formulations and their respective carriers to induce an inflammatory response. In vaccine development, engineered nanomaterials are often even intentionally used to enhance or trigger an inflammatory response linked to cytokine expression such as in the case of cancer immunotherapy [61,62,63]. However, the intentional induction of inflammation by engineered nanomaterials is a multifaceted area that requires specific attention to the potential benefits and risks, in particular, their potential long-term consequences. Overall, the safety and potential health effects of nanomedicine products are an area of active research and debate. Various studies suggested that some nanomedicine products, such as liposomal drug carriers, nanoparticles, and dendrimers, trigger immune stimulation with health-threatening outcomes in some cases [27,47,64]. The panel of cytokines selected in our study serves as a model for the main response of immune cells in human blood. The components of the panel include potent inflammatory factors such as TNF-α, IL-1Ra, IL-1β, IL-6, IFN-γ, and GM-CSF involved in the pathogenesis of various inflammatory and autoimmune diseases, cytokines with activating/homeostatic functions produced by lymphocytes such as IL-2, anti-inflammatory factors, and factors involved in type 2 alternative inflammation such as IL-4 and IL-10. Our data show that for both TNF-α and IL-10, the free drug seems to show as strong or stronger effect than the Dox2 formulation. In case of TNF-α and IL-8, the contribution of the liposomal carrier itself can be also observed, while elevated amounts of IL-2 and IL-4 seem to be linked to the presence of a loaded carrier or endotoxin contamination.

These findings highlight the importance of rigorous quality control regarding the free drug content in liposomal drug products, not only because of a possible straightforward cytotoxic effect but also because of unexpected interactions of lipophilic drugs with serum components participating in immune-response processes.

We have shown that the combination of assays presented here—including physico-chemical characterization, in vitro cytotoxicity, complement activation, and cytokine profile—could be considered a preliminary pre-screening step for the characterization of complex heterogeneous formulations, like Dox2, and is able to highlight potential problems in the formulation, prior to arriving at non-clinical biodistribution studies or clinical evaluation. In particular: (i) to assess some of the equivalent liposome characteristics, e.g., the state of the encapsulated drug, liposome size distribution, and in vitro leakage rates; (ii) more importantly, to anticipate differences and biological effects at early stages that may otherwise be detected in non-clinical biodistribution assays, and/or in clinical evaluation currently required to access biosimilarity.

With this work, we want to emphasize that building a comprehensive knowledge base generated during manufacturing, analysis, and material control on the physico-chemical and biological characteristics to better understand potential risk is extremely important. This knowledge can be gained from the early stages of pharmaceutical research and development and has to be updated in the case of subsequent manufacturing and associated control strategy over time.

## 4. Materials and Methods

### 4.1. Selection of Test Materials

Two formulations of Doxil and their respective carrier controls were assessed in this study as representative test items. Doxorubicin hydrochloride (Cat 44583, Sigma, Milan, Italy) was also included in our in vitro and ex vivo experiments for comparison purposes. The Doxil formulation, here named Dox1, and its drug-free carrier control, referred to by the acronym Dox1C, were purchased from Producer 1, while the Dox2 formulation and its corresponding carrier control, named Dox2C, were purchased from Producer 2. Samples were stored frozen in their original form (in 10% sucrose and 10 mmol/L histidine buffer) at −20 °C.

Due to the nature of the research, more information on the products can be obtained only on request.

### 4.2. Physico-Chemical Characterization

Dynamic light scattering (DLS) in batch mode was applied to determine particle size distribution (PSD) and polydispersity index (PDI) of the samples, using a Zetasizer Nano-ZS (Malvern Instruments Ltd.; Malvern, UK). Samples were diluted 50 times (*V*/*V*) in phosphate buffered saline solution and 0.5 mL of the resulting suspension was loaded in disposable semi-micro PMMA cuvettes. Measurements were performed three times for each sample at 25 °C after 120 sec equilibration time, applying automatic measurement duration and automatic seek for optimal position options. The automatically selected attenuator value was 6 for all samples. Data were evaluated by Zetasizer software version 7.12 (Malvern Instruments Ltd., Malvern, UK). A material refractive index of 1.47, material absorption of 0.001, dispersant refractive index of 1.334, and viscosity of 0.911 cP were applied during data analysis in “general purpose” mode. Zeta average, PDI, and major peak mode values were extracted from the average of 3 measurements. In order to describe size variability, FWHM values were calculated by plotting the average intensity-based distributions on a linear scale as the size distributions were not symmetrical, but tailed toward larger size values, as observed also in the case of other nanoparticulate suspensions [34].

A flow-field-flow fractionation (AF4) system (AF2000; Postnova Analytics, Landsberg, Germany), coupled with a UV–vis detector and a Zetasizer Nano-ZS (Malvern Instruments Ltd.) DLS detector was used to refine information on the hydrodynamic diameter of the loaded and carrier liposomes. A trapezoid-shaped channel with 275 mm length and 350 µm thickness was applied, combined with a 10 kDa cut-off regenerated cellulose membrane. Liposomes were diluted in filtered PBS before injecting them in the filtered PBS mobile phase applying a 20 µL loop. A detector flow of 0.5 mL/min, injection flow of 0.2 mL/min, crossflow of 1 mL/min and focus flow of 1.3 mL/min were applied during the 10 min injection phase. The focus flow was decreased to 0 using 1 min transition time and the cross-flow was kept for an additional 0.5 min at 1 mL/min. Then, the crossflow was decreased to 0 in 60 min following a power-function-type decay with exponent of 0.5. An additional 30 min elution time was added at 0 cross-flow in order to observe the possible elution of larger aggregates. The UV detector was set to 280 nm, while DLS measurements were run at fixed attenuator value of 11 at 3 s run durations. ZEN0023 type flow cell and “centre of the cell” type positioning was selected. Data were analysed utilising the AF200 Control software version 1.1.1.26 (Postnova Analytics, Salt Lake City, UT, USA) and the Zetasizer software version 7.12 (“General purpose” mode). Average hydrodynamic diameter D_FWHM_ values were calculated from the size values corresponding to the FWHM time points.

Stokes diameter and free drug content of the samples was determined using a Beckman Coulter ProteomeLabTM XL-I analytical ultracentrifuge (AUC) (Beckman Coulter Inc., Brea, CA, USA). Liposomes were 50 times diluted in filtered (0.22 µm) PBS and the same buffer solution was used as control in the reference cell. Two sector, sapphire window cells were loaded with 390 µL sample and 400 µL reference buffer (in separate sectors) and placed in an 8-hole Ti rotor. Measurements were run at 10,000 rpm rotational speed at 20 °C nominal temperature. Data were collected overnight (1120 min for loaded and 977 min for control liposomes) using interference and absorbance optics for loaded liposomes, and only interference optics with a 2 min pause between measurements for control liposomes. As the absorption maximum of doxorubicin changes slightly due to encapsulation, detection wavelength of 514 nm—where free and encapsulated doxorubicin provided the same optical signal at the same theoretical concentration—was selected for absorbance measurements based on a former study [37]. The ls-g*(s) model of the Sedfit software v.15.01b 22–24] was applied to fit interference data (using linear grid with a resolution of 200 in the 1-800 S range, selecting Baseline, Fit RI Noise, Fit Time independent noise and Meniscus, confidence level 0.95) and to calculate sedimentation coefficient distributions, which were transformed into mass-based size distributions using the same software (using the ‘transform s distribution to r distribution’ option), applying an estimated effective density of 1.055 g/mL for loaded and 1.045 g/mL for unloaded carrier liposomes (determined in former studies [17]). The density and viscosity of PBS at 20 °C was considered to be 1.0056 g/mL and 0.01022 mPa·s, respectively [65]. The contribution of free doxorubicin was estimated based on absorbance measurements following the method described earlier [37].

### 4.3. Assessment of Endotoxin Contamination

The endotoxin contamination was measured using the Limulus amoebocyte lysate (LAL) chromogenic methods by kinetic assay. The standard endotoxin from the *E. coli* O111:B4 strain (cat. EC010, 10 ng/vial; Associates of Cape Code, Inc., East Falmouth, MA, USA) was used to prepare the control standard curve (CSE). All dilutions were made in endotoxin-free water (LAL Reagent Water, cat. WP1001; Associates of Cape Code, Inc.). The glucashield^®^ Reconstitution Buffer (cat. CG1500; Associates of Cape Cod, Inc., East Falmouth, MA, USA) was used for LAL reconstitution.

Standard endotoxin (0.5 EU/mL) was “spiked” in the nanopharmaceutical samples to detect possible interference with the assay. The Pyros Kinetix^®^ Flex Tube Reader (Associates of Cape Code, Inc.) was used to acquire sample readouts at 405 nm, according to manufacturer’s instructions. The raw data were analysed with Pyros^®^ EQS software v1.2. Results for each individual sample were considered valid only if the correlation coefficient of the calibration curve was ≥0.98, and if the recovery rate of spiked controls was within 50% and 200% according to US FDA guidelines [66,67,68]. The assay sensitivity was 0.001 EU/mL.

For the presented immunological studies, the tested materials have been further diluted (see the European Pharmacopoeia acceptance value for injectable drugs of 0.25 EU/mL below) to reach a final endotoxin level of 0.18 EU and 0.14 EU for Dox2 and Dox2C, respectively.

### 4.4. Cytotoxicity Evaluation

The in vitro cytotoxicity study of the selected nano-pharmaceuticals was evaluated with colorimetric MTT assays in LLCPK1 for 48 h exposure according to the EU-NCL-GTA02 SOPs [69]. The cleavage of the chromogenic substrate MTT correlates with the cell metabolic activity and therefore with the number of living cells.

Cells of the human hepatocellular carcinoma line LLCPK1 were plated in 96-well cell culture plates (Corning Inc., Corning, NY, USA) at a density of 2.5 × 10^4^ cells/well and allowed to adhere for 24 h, then exposed to the liposomes (1 to 50 μg/mL doxorubicin concentration) for 48 h. A medium control and a positive control (Triton 0.1%) were included in each assay. Cell viability was evaluated using MTT [3-(4,5-dimethylthiazol-2-yl)-2,5-diphenyl-2H tetrazolium bromide] (Sigma-Aldrich, Inc., St. Louis, MO, USA) added to the cells in fresh complete culture medium at a final concentration of 250 μg/mL. After 4 h of incubation at 37 °C, the supernatant was removed, and the precipitated formazan crystals (indicative of mitochondrial metabolic activity, i.e., presence of viable cells) were dissolved in 200 µL DMSO (Sigma-Aldrich, Inc.) followed by 50 µL of glycin buffer (0.1 M glycine with 0.1 M NaCl). The absorbance was quantitated at 570 nm by the EnSpire^®^ Multimode plate reader (Perkin Elmer) using a reference wavelength of 680 nm. Data are expressed as percent of mitochondrial activity in respect to the negative control, and reported as mean ± SD. Three independent experiments were performed in triplicates. Statistical analysis is reported in Section 4.7.

### 4.5. Complement Activation Assay

The effect of both Doxil formulations, their vehicles, and free doxorubicin (DoxHCl) on complement activation was measured in commercially available human pooled serum (cat. cod A113, Quidel, Santa Clara, CA, USA). AUC measurements revealed that about 13% of the doxorubicin content of the Dox2 product is present in free form, outside the liposomes. Therefore, the capacity of free doxorubicin hydrochloride (DoxHCl) to raise the levels of iC3b was also evaluated at 10 and 20% of the theoretical load of the nanoformulation (at the applied 200 µg/mL doxorubicin concentration). The applied concentration was selected to assess the effects of acute exposure, given the difficulties of conducting a reliable chronic exposure test of these nanoformulations and considering a typical concentration of 20 µg/mL in blood. In order to confirm the concentration-dependent effect of DoxHCl, human pooled serum was also exposed to DoxHCl concentrations ranging from 2.5 to 20% of the total theoretical load.

The selected materials were prepared in sterile PBS mixed with human serum 1:5 (*v*/*v*). PBS and Kolliphor EL (CAS 61791-12-6); Sigma Aldrich) were included as negative and positive controls, respectively. Samples were incubated in a final volume of 250 µL for 1 h at 37 °C under orbital shaking at 300 rpm. Then, 20 µL of EDTA 200 mM, pH 8.0, was added to block the complement reaction. The generation of the complement cascade cleavage product iC3b, was tested with a commercial enzyme-linked immunosorbent assay (ELISA) kit (cat. A006, Quidel), according to the manufacturer’s instructions, after appropriate dilution in specimen diluent (1:100 for iC3b). The presence of iC3b was measured with an EnSpire^®^ Multimode plate reader (Perkin Elmer) at 405 nm. Two independent assays were performed, and technical duplicates were run in the ELISA.

The manufacturer’s protocol has been followed to calculate iC3b concentration in the sample specimens, referring to assigned concentration of the standards and the control vials (available in the certificate of analysis) expressed as absolute units of iC3b protein and taking into account the specimen dilution made.

The slope, intercept, and correlation coefficient of the derived best-fit line for the iC3b A, B, and C standards has been calculated and values were within the specified ranges to qualify the assay (correlation coefficient (r): >0.95; slope (m): between 0.72 and 1.30; y-intercept: between 0.70 and 1.39).

### 4.6. Inflammation-Related Cytokines

The inflammatory effects of both Doxil formulations, their vehicles, and doxorubicin was assessed using the same set of human serum samples used in the complement-activation measurements. Similarly to those experiments, the effect of free DoxHCl was also studied at 10 and 20% of the theoretical DoxHCl load of the formulations.

In brief, human serum incubated with only PBS was used as negative control, while human serum incubated with LPS (*E. coli* O111:B4, cat. 00-4976; Invitrogen, Thermo-Fisher Scientific, Waltham, MA, USA) at 10 ng/mL final concentration was used as positive control. The experiment was run with two technical replicates.

Samples were thawed on ice and tested for the presence of cytokines using commercial ELISA-based microarrays that simultaneously measure multiple proteins in a single sample aliquot.

A Multiplex Bio-Plex Pro™ Human Cytokine 8-plex Assay (cat. M50000007A) was used for assessing the production of IL-2, IL-4, IL-6, IL-8, IL-10, TNF-α, IFN-γ, and GM-CSF. Singleplex for IL-1β (Cat 171B5001M) and IL-1-Ra (Cat 171B5002M) were also included. Samples were run according to the manufacturer’s instructions. An internal calibrator was used. Cytokines were analysed with the Bio-Plex200 System using the Bio-Plex Manager^TM^ software, and data were analysed by the Bio-Plex Data Pro^TM^ software (v 6.1.1), using five-parametric curve fitting. For each cytokine, assay ranges and LOD were provided by the manufacturer. All reagents and instruments, including Washing Station and Shaker Incubator, were from BIO-RAD Laboratories, Inc., Hercules, CA, USA.

### 4.7. Statistical Analysis

Graphs were plotted using OriginLab 2022b software and the statistical analysis was performed using GraphPad Prism version 5.03 for Windows (GraphPad Software, La Jolla, CA, USA Software). Data are reported as mean results of technical replicates ± SD. Complement-activation and inflammatory-response significance were assessed by one-way analysis of variance (ANOVA), followed by Dunnett’s multiple comparison post hoc test. Differences were considered as statistically significant among sample groups when *p* < 0.05.

## 5. Conclusions

Although, in recent years, the area of lipid-based nanomedicines has seen enormous progress due to improvements in the technology for industrial production, potential health risk(s) originating from errors in manufacturing processes still happen. Quality control analysis, including not only control of ingredient materials, but also the rigorous analysis of physico-chemical properties, are of outmost importance as they correlate with the biological characteristics and therefore safety and efficacy of the formulation itself. Our representative examples illustrate the relevance of a proper characterization approach in the specific case of nanosimilars. However, it serves also as general example in demonstrating that the application of a correct characterization strategy has key importance in product-safety evaluation and the identification of possible health threats due to inappropriate product properties.

The present research, therefore, contributes to a growing body of evidence suggesting the need for assessing the safety of nano-drug formulation through systematic studies that combine a robust physico-chemical characterization and a nano-drug specific testing battery to also assess the function of the immune system, such as complement activation and inflammation. Moreover, proper physico-chemical characterization at the product development phase as part of the recently often-quoted “Safe by Design” approach is of paramount importance in the case of nanomedicines, as the efficacy for which these formulations are designed, their hazardous profile, and the effects they might have due to them are strongly linked to their physico-chemical properties.

## Figures and Tables

**Figure 1 ijms-24-13612-f001:**
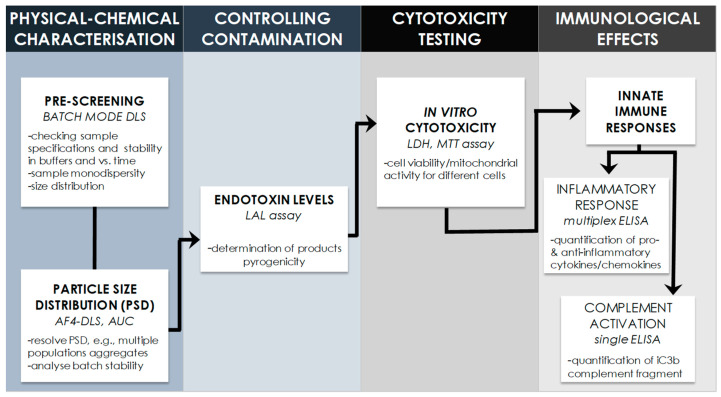
Each Doxil formulation and its corresponding carrier were characterised for the physico-chemical properties, the potential endotoxin contamination, then examined for their potential toxicity on human cells and capacity to activate innate immune responses in human serum. The figure above describes the multi-step characterisation strategy applied in this study.

**Figure 2 ijms-24-13612-f002:**
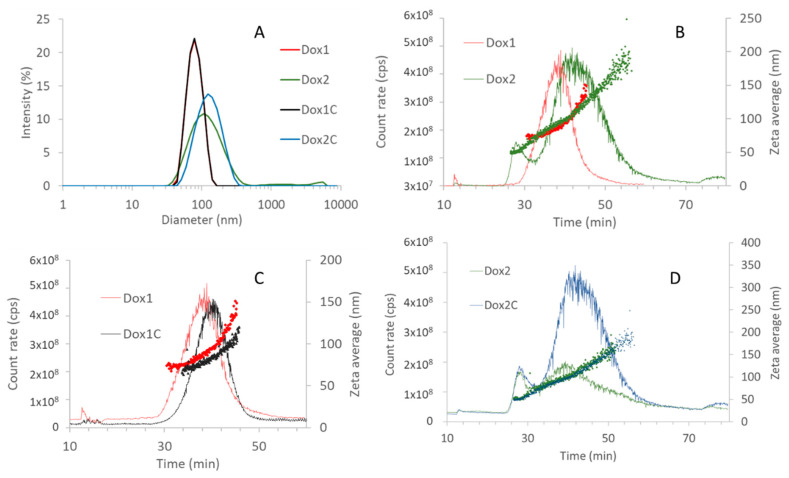
(**A**) Batch-mode DLS data reported as intensity-based differential size distribution. Averages of three measurements are shown. AF4-DLS analysis results of (**B**) Dox1 and Dox2 loaded liposomes and (**C**) Dox1 and its corresponding carrier formulation, (**D**) Dox2 and its corresponding carrier formulation in PBS. The scattered intensity and size vs. elution time from flow mode-DLS data are shown for a representative measurement for each selected sample for Dox1 in red, Dox1C in black, Dox2 in green, Dox2C in blue. Hydrodynamic diameters are represented with dots, scattering intensity peaks with lines. Details of experimental conditions are reported in Section 4.2.

**Figure 3 ijms-24-13612-f003:**
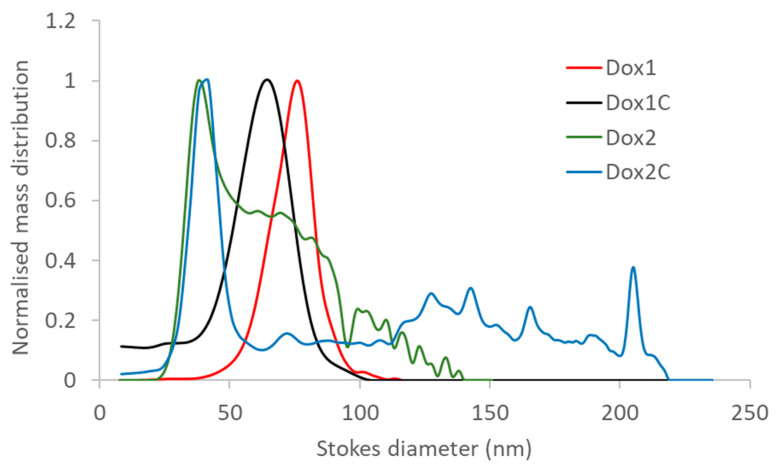
Size distributions (Stokes diameter) of the loaded Dox1 and Dox 2 and their corresponding carrier formulations determined from AUC measurements.

**Figure 4 ijms-24-13612-f004:**
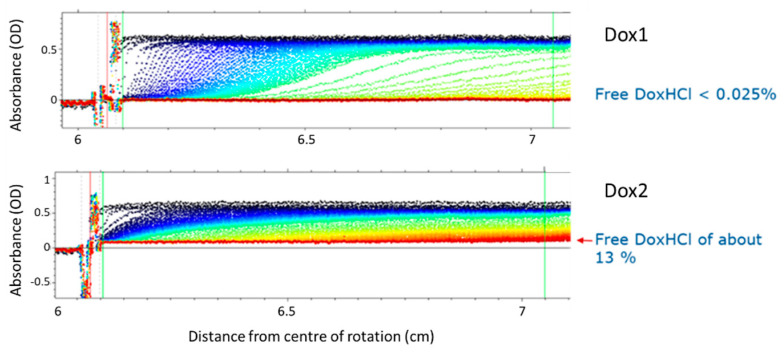
AUC absorbance measurement raw data indicate non-sedimenting fraction: free doxorubicin hydrochloride (DoxHCl). Negligible contribution in Dox1, about 13% in the Dox2 formulation (bottom). Details of experimental conditions and reference to calculation method are reported in Section 4.2. Various colours represent various time points (from start to end colour changes from dark blue to red.)

**Figure 5 ijms-24-13612-f005:**
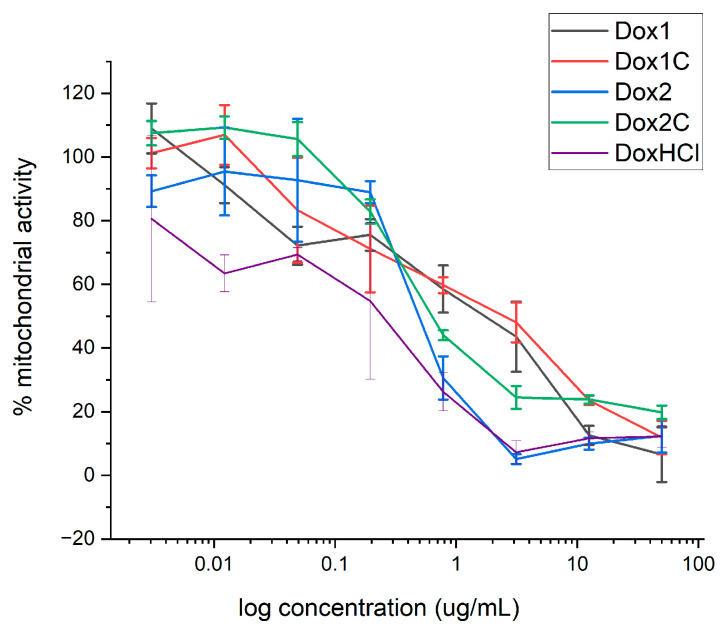
Dose–response curves (% mitochondrial activity (MTT)) of LLCPK1 cells exposed to the two Doxil nanoformulations, their controls, and free DoxHCl for 48 h. The error bars represent the SD of three independent experiments. Negative controls (incubation in medium) and positive controls (cell lysis with Triton) were used as benchmarks.

**Figure 6 ijms-24-13612-f006:**
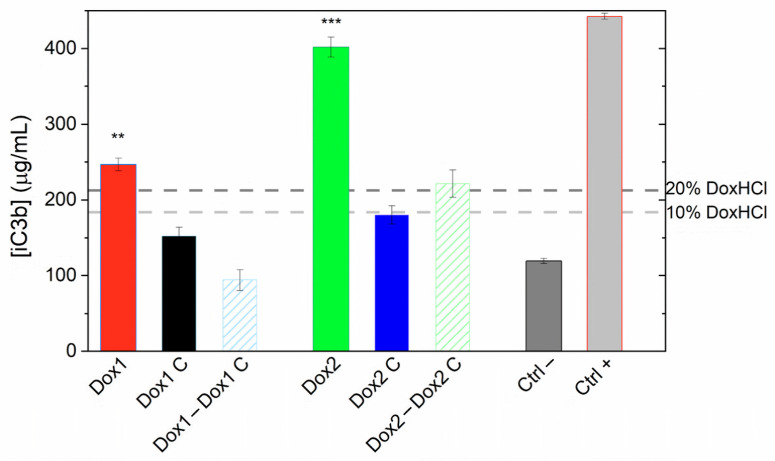
Levels of iC3b measured in human pooled serum upon incubation with the two liposomal formulations (Dox1 and Dox2) and their carriers (Dox1C, Dox2C) at 200 µg/mL for 1 h at 37 °C. PBS and Kolliphor 0.5% (*v*/*v*) were used as negative and positive controls, respectively. Data are shown as mean values ± SD of two independent experiments. The dotted horizontal lines show the values of iC3b amount generated by free doxorubicin (DoxHCl) at concentrations corresponding to 10% and 20% of the total load in 200 µg/mL liposomal formulation. ** *p* < 0.01, *** *p* < 0.001.

**Figure 7 ijms-24-13612-f007:**
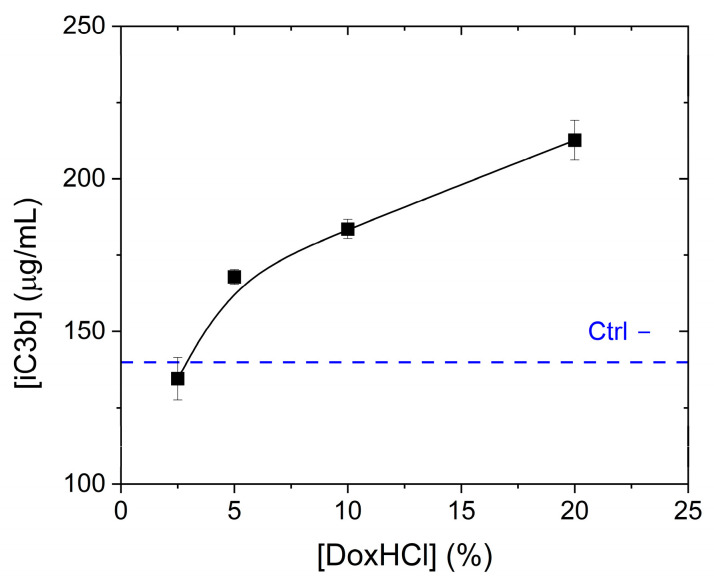
Levels of iC3b measured in human pooled serum upon incubation with free doxorubicin hydrochloride at various concentrations ranging from 2.5 to 20% of the total theoretical load at 200 µg/mL doxorubicin concentration, for 1 h at 37 °C. Data are shown as mean values ± SD of two independent experiments. The iC3b level of the negative control (PBS) is indicated as horizontal blue dotted line.

**Figure 8 ijms-24-13612-f008:**
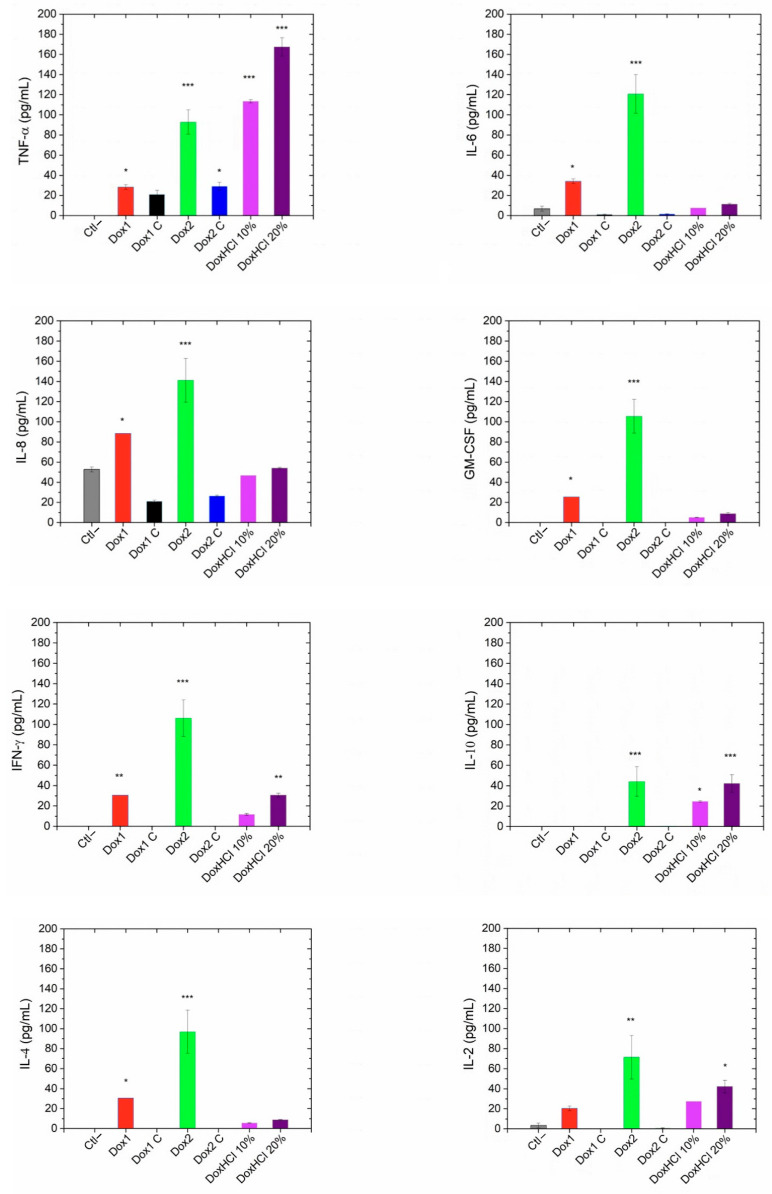
Human pooled serum exposed to Dox1 (red), Dox2 (green), their corresponding carriers Dox1C and Dox2C (black and blue) as well as free doxorubicin hydrochlorate tested at 10 and 20%, DoxHCl (light and dark purple) after 1 h incubation at 37 °C. The data of a panel of cytokines/chemokines (TNF-α, IL-6, IL-8, GM-CSF, IFN-γ, IL-10, IL-4, IL-2) is represented here. Untreated samples are indicated as Ctl−, whereas LPS (10 ng/mL) was used as a positive control and worked with no change in cytokine/chemokine levels as expected in serum. Data are also shown as mean ± SD. * *p* < 0.05, ** *p* < 0.01, *** *p* < 0.001.

**Figure 9 ijms-24-13612-f009:**
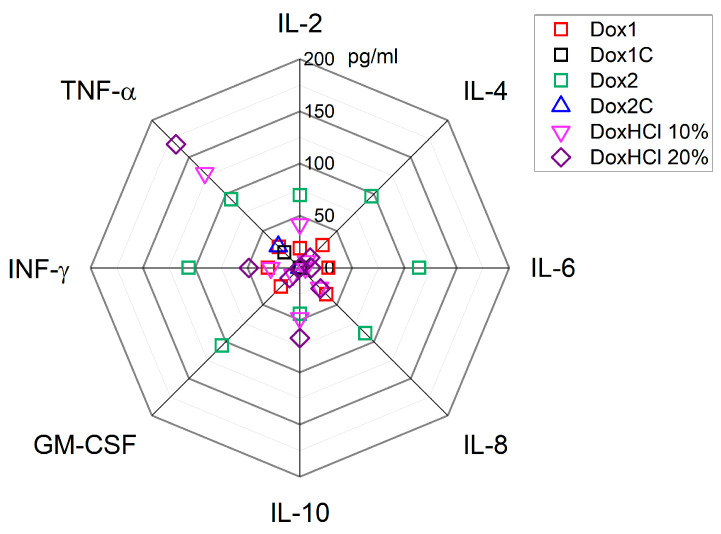
Radar plot for a panel of 8 cytokines/chemokines (IL-2, IL-4, IL-6, IL-8, IL-10, TNF-α, GM-CSF, IFN-γ) measured in human pooled serum exposed to Dox1, Dox2, their carriers (Dox1C, Dox2C) and free doxorubicin 10 or 20% DoxHCl (% of theoretical liposome load) after 1 h incubation at 37 °C. Data are represented after subtraction of the negative control value.

**Table 1 ijms-24-13612-t001:** Hydrodynamic diameter, PDI, mode of intensity based peak and full with at half maximum (FWHM) of the major peak of the two liposomal doxorubicin formulations and their corresponding non-loaded carriers determined by batch-mode DLS.

Liposome	z-Average (nm) ±Standard Deviation	PDI	Major Peak Size Dh (nm)	FWHM (nm)
Dox1	77 ± 0.9	0.033	81	54
Dox1C	77 ± 0.4	0.038	81	53
Dox2	107 ± 1.2	0.234	128	154
Dox2C	116 ± 0.6	0.122	134	138

**Table 2 ijms-24-13612-t002:** Endotoxin contamination of tested materials measured by chromogenic LAL test. Endotoxin contamination levels are expressed in EU/mL.

Materials	LAL Test (OD_540nm_)	
	Endotoxin (EU/mL)	Recovery Rate (%)
Dox1	not detected	104%
Dox1C	not detected	100.6%
Dox 2	4.10 EU/mL	105%
Dox2C	3.01 EU/mL	92.9%

## Data Availability

Not applicable.

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
