# Peer review of "Differences in Physico-Chemical Properties and Immunological Response in Nanosimilar Complex Drugs: The Case of Liposomal Doxorubicin"

_ijms, 2023, doi:10.3390/ijms241713612_

Round 1
Reviewer 1 Report
The present submission is concerned, by title, with the physico-chemical characterization of nanocarriers exemplifying the case of liposomal doxorubicin, an objective of study dealt with in a multiplicity of studies of the authors. While this “characterization” topic in nanomedicine is certainly important, there are several issues regarding novelty and contextualization of the present study. In instances, no experimental evidence for the provided claims is provided. Also, important citations are missing and the authors exclusively move in the context of their own work, which should be addressed. Though such critique may appear harsh, it is to strengthen the scientific objective of study including a proper scientific discourse.
1. The experimental section regarding the characterization, particularly section 4.2. is in an unacceptable shape and does not allow any experiment or analysis to be repeated in an appropriate fashion.
Details of DLS and data analysis are missing.
The description of field-flow fractionation does not contain any elution conditions, utilized membranes, focusing, elution strategies……
The description of AUC does not allow any repetition of experiments. Experimental conditions, data acquisition, and data analysis details are not provided.
In fact, all those techniques have been studied in a comparative fashion in a recent study by Cinar et al. (Analytica Chimica Acta, 2022, 1205, 339741) including an in-depth discussion of the pros and cons. Some of the findings agree and some disagree with the here provided results, which is interesting.
2. The abstract mentions Covid 19 and vaccines, though the submission does not include material associated to such application nor any discussion related to those objectives of study based on experimental data.
3. Line 48: Pegylated lipids, especially those of the vaccines have been studied in depth in a detailed investigation by Anufriev et al. (Analytical Chemistry, 2023, 95, 10795–10802.)
4. I would be cautious with a statement such as on lines 54 to 56. If PEGs are so tremendously “cumbersome”, why a significant fraction of the worlds entire population could have been vaccinated successfully against Sars-Cov-II? I assume such high percentage of them being affected through such issues would have caused some public attention. Anuvriev et al. also discusses the importance of polymer alternatives to PEG because of its wide use. Some literature discusses this topic more controversally.
5. Figure 1 in Section 2 is not introduced nor set in context with enough description of its purpose and relation among the different aspects. This should be improved.
6. Table 1 and, e.g., lines 90 to 91. The PDI alone is not decisive for homogeneity of products. The authors can easily calculate the variation of sizes from DLS data that sets into context the PDI and absolute size (Analytica Chimica Acta, 2022, 1205, 339741). The size variation is particularly more useful since the PDI value by itself results in different size ranges implied by the difference in absolute values of size.
7. Section 2.2 dismisses details and should outline which size (from the range mentioned) is the “appropriate”. Again, the authors can find information on that in a recent study (Analytica Chimica Acta, 2022, 1205, 339741) apart from their own work. Why only a size range and not an appropriate size value from this range is calculated as a figure of merit? It might help to rationalize some aspects reported here.
8. Section 2.3. claims similarity of results between DLS and AF4. This is not always the case (e.g., Analytica Chimica Acta, 2022, 1205, 339741). The way the size distributions are created should be mentioned. Is this a differential distribution of sizes? If, how has it been derived from differential distributions of sedimentation coefficients the authors determined? What makes it a mass-based distribution?
Which formula the Stokes size is related to? It has been shown recently that micellar structures and hydrated objects in solution must consider the translational friction properties due to hydration. May consideration of this aspect change the results significantly? (see, e.g.: Analytical Chemistry, 2023, 95, 10795–10802; Analytical Chemistry, 2021, 93, 15805-15815.) I miss information on how the partial specific volume necessary to calculate the sizes was determined. It would be useful for the reader to be reported.
9. Lines 137-138: The authors mention a different diameter of “empty” and “loaded” liposomes. This may simply be associated to differences in density of the objects and an artifact by the (erroneaously determined / assumed) partial specific volumes. Has the partial specific volume been determined here? Maybe it also changes depending on load with the drug doxorubicin as has been show recently by Demeler et al. for nonrelated lipid nanoparticle formulations.
10. Lines 139-141: The authors report on full agreement of AF4-DLS and AUC results. That is really surprising and should be substantiated in view of recent results on nanomedical particle systems including commercial polymers (Analytica Chimica Acta, 2022, 1205, 339741). It would be useful if average sizes, being representative of the data, are calculated.
11. Line 145: Why would a large amount of particles with a low diameter be unexpected? This is a well-known fact of proper characterization work that needs to be referenced also with work apart from the authors own.
12. Figure 4 is not discussed in detail enough. There has been a detailed study on nanoparticle formulation analysis that considered the presence of free and encapsulated drugs (Analytical Chemistry 2020, 92, 7932-7939.) For the quantification, the UV response of drug in the particle and outside needs to be checked for, including linearity of detectors against concentration in each case. The amount of free DOX in DOX 1 appears to indicate an accuracy impossible to reach at the signal-to-noise ratio in the data.
13. The biochemically-associated characterization appears okay to me, except that details are missing of how the significance of the results was checked for. As for the technical characterization part, a more in-depth discussion of the experiments and obtained results would be desirable.
14. Lines 294-296 refer to work from the authors own institution only and do not imply any novelty that, in addition, neglects the body of work from other authors. Why? It would enable a much wider and cooperative view of science performed by many research groups worldwide.
15. Lines 297-305 just repeat results without substantiation by a proper discussion. The sizes and their variation can be compared and will beautifully confirm the authors conclusions by the respectively calculated data.
16. Lines 313-314: The ultimately determined and calculated hydrodynamic diameter requires substantiation and is not mentioned in the present work. Indeed it is well-known that the separation according to diffusion coefficients/size and light scattering on elution fractions significantly improves insight on the sample.
17. Lines 320-323: It would be desirable to clarify why the sizes are mass-based and which nature the distributions have. The calculation of the size from the distributions should be outlined or discussed. Was it the first statistical moment? (Analytica Chimica Acta, 2022, 1205, 339741) The linear correlation between particle concentration and refractive index change is true also for UV detection under conditions where the Mie scattering is not significant as adressed, e.g., by Cölfen et al.
18. Lines 323-330 are not substantiated and evidenced by the experiments in which only a single sedimentation velocity run is performed. The experimental section describes experiments with a refractive index detector only, though UV data are presented in Figure 4. No cell materials, no data acquisition, etc. are described. Essentially, this is going back to comment 12 where this issue of UV responses was mentioned.
19. Lines 338-340 are very vague. Is it released doxorubicin or was it not encapsulated originally? The many sources of the observation need to be rationalized by something most likely. Has stability of those dispersions be studied or addressed?
20. Lines 367-368: The authors stress accurate characterization at all levels, however, should underline it with appropriate results. At present, the here presented experimental description and results are difficult to justify such claim.
21. Lines 439-443 mention the authors strategy the “correct characterization strategy” which is, in view of the here reported results, quite confident. The Dox Liposomes have been studied multiple times by Mehn et al. with limited new information reported in the present study and very similar conclusions from a single AUC sedimentation velocity run.
22. Throughout the manuscript, the consistency of wording should be checked. There is, e.g., a mix between “physical-chemical” and “physico-chemical” throughout the entire manuscript.
Overall, the language of the manuscript appears appropriate. In instances it would be useful if the consitency of the language would be checked for.
Reviewer 2 Report
The work by Lipsa et al presents a very robust framework for the assessement of the physico-chemical properties and in vitro toxicity of liposomal doxorubicin formulations that could be considered a pre-screening step for assessing formulation changes in the context of CMC and also for the assessment of biosimilarity. The results are well argumented. The strong interest lies in the combination or orthogonal and complementary methods including the correlation of physical-chemical characterisation methods, complement activation and in vitro toxicity experiments as a robust approach to detect and anticipate failures that may be expected in vivo non clinical biodistrubution studies or clinical studies.
I suggest to publish the manuscript after a minor revision of the introduction and of the conclusions
Introduction:
Considering that the manuscript focuses on liposomes and not on RNA-LNPs I would suggest to consider revising the introduction to include a few more words on what physico-chemical parameters, pharmacokinetics, ect, etc that was considered in the past to assess the biosimilarity of generic doxorubicin formulations (e.g. Sun Pharma's generic doxorubicin HCl liposome injection versus Caelyx®). Some information may be available at EMA (https://www.ema.europa.eu/en/medicines/human/withdrawn-applications/doxorubicin-sun) or FDA websites (https://www.accessdata.fda.gov/drugsatfda_docs/psg/Doxorubicin%20Hydrochloride_draft_Injection%20injec%20lipo_RLD%2050718_RC09-18.pdf). it may also worth mentioning the main attributes and studies required in the dedicated guidance documents published by FDA and EMA, and how the presented measurements methods related to those documents.
All the references to LNPs-RNA may not be needed unless shared needs between liposomal formulations and LNPs justify the discussion. If is the purpose of the authors in the introduction is also to expand the discussion beyond liposomal doxorubicin, other formulations may be considered, including iron sucrose NBCDs.
Discussion:
In the discussion and the conclusions it would be interesting to better explain how the combination of the assays presented in this work including the physical-chemical characterisation, cytotoxicity and complement activation in vitro could be considered a preliminary pre-screening step for the characterisation of complex heterogeneous formulation, like the Dox2 prior to perform non clinical biodistribution studies or in clinical evaluation of the formulation. In particular:
i) to assess some of the equivalent liposome characteristics, e.g. state of encapsulated drug, liposome size distribution, and in vitro leakage rates….
but also (and possibly more importantly)
i) to anticipate at early stages differences and biological effects that may otherwise be detected in non-clinical biodistribution assays, and/or in clinical evaluation currently required to assess biosimilarity.
An useful reference for expanding the reflection beyond the ones already cited may be: 10.5639/gabij.2014.0302.017.
the quality of the english language seems acceptable
Round 2
Reviewer 1 Report
I reread the entire manuscript and have to admit that the authors made valid points in their discussion of the material. The changed title appears much more suitable for the study, the introduction provides much more context. All experiments are described to as much detail as possible. The cited literature on PEG and its issues provide a modified picture for its problems, which I thought were less critical. In instances the answer to comments helped me to reflect my viewpoint. A very small comment: On line 579 the unit of viscosity is missing.
I do not see any issues with the language that appears accurate to me.